# Human Genes Involved in the Interaction between Host and Gut Microbiome: Regulation and Pathogenic Mechanisms

**DOI:** 10.3390/genes14040857

**Published:** 2023-03-31

**Authors:** Luigi Boccuto, Jan Tack, Gianluca Ianiro, Ludovico Abenavoli, Emidio Scarpellini

**Affiliations:** 1School of Nursing, Healthcare Genetics Program, Clemson University, Clemson University School of Health Research, Clemson, SC 29631, USA; 2Translational Research Center for Gastrointestinal Disorders (T.A.R.G.I.D.), Gasthuisberg University Hospital, KU Leuven, Herestraat 49, 3000 Lueven, Belgium; 3Department of Medical and Surgical Sciences, Digestive Disease Center, Fondazione Policlinico Universitario Agostino Gemelli IRCCS, 00168 Rome, Italy; 4Department of Translational Medicine and Surgery, Università Cattolica del Sacro Cuore, 00168 Rome, Italy; 5Department of Health Sciences, Magna Graecia University, 88100 Catanzaro, Italy; 6Clinical Nutrition and Hepatology Unit, San Benedetto del Tronto General Hospital, 63074 San Benedetto del Tronto, Italy

**Keywords:** gut microbiota, gut microbiome, metagenomics, immune system, obesity

## Abstract

Introduction: The umbrella term “human gut microbiota” describes the complex ecosystem harboring our gut. It includes bacteria, viruses, protozoa, archaea, fungi, and yeasts. This taxonomic classification does not describe its functions, which encompass nutrients digestion and absorption, immune system regulation, and host metabolism. “Gut microbiome” indicates instead the genome belonging to these “microbes” actively involved in these functions. However, the interaction between the host genome and the microbial ones determines the fine functioning of our organism. Methods: We reviewed the data available in the scientific literature on the definition of gut microbiota, gut microbiome, and the data on human genes involved in the interaction with the latter. We consulted the main medical databases using the following keywords, acronyms, and their associations: gut microbiota, gut microbiome, human genes, immune function, and metabolism. Results: Candidate human genes encoding enzymes, inflammatory cytokines, and proteins show similarity with those included in the gut microbiome. These findings have become available through newer artificial intelligence (AI) algorithms allowing big data analysis. From an evolutionary point of view, these pieces of evidence explain the strict and sophisticated interaction at the basis of human metabolism and immunity regulation in humans. They unravel more and more physiopathologic pathways included in human health and disease. Discussion: Several lines of evidence also obtained through big data analysis support the bi-directional role of gut microbiome and human genome in host metabolism and immune system regulation.

## 1. Introduction

The gut microbiota can be considered a complex ecosystem encompassing bacteria, viruses, protozoa, archaea, yeasts, and fungi harboring in our intestine [1]. Its composition changes just like our genetic, metabolic, and physiological activities fluctuate during life [2]. Our gut microbiota reach stable configuration at the host age of 2–3 years, conferring to every human being a peculiar microbial “passport” characterized by a relative abundance of original microbial strains [3,4]. In general, all human beings have gut microbiota whose main phyla belong to *Bacteroidetes* and *Firmicutes* [5]. Specifically, more than 50 bacterial species are shared by all individuals belonging to the human species [6]. Thus, there is a corresponding functional bacterial core preserved from extinction by a common human gut metagenome. The latter can be defined as “the genome reconstructed through the application of modern genomics technique without the need for isolation and lab cultivation of individual species”. Metagenome is required for the correct functioning of gut microbiota and intestine, which is considered part of a perfect ecosystem. The functions encoded in this core functional minimal human and bacterial gut metagenome include regulation of the host–microbiome interactions, nutrient absorption, and metabolism (e.g., degradation of complex polysaccharides and synthesis of short-chain fatty acids, considered the fuel for enterocytes growth and maturation) [7].

The composition of the gut metagenome is highly relevant for human health and disease. In the case of energy homeostasis, a subset of human genes coordinates these metabolic processes. These genes cannot explain the high “personal” variability in energy homeostasis maintenance among humans. Interestingly, several pieces of evidence from the literature have shown that gut microbiota are able to affect energy balance. For example, our intestinal microbes can dynamically modulate the efficiency of calorie harvesting from ingested food [8,9]. In detail, obese subjects have a lower intestinal bacterial diversity and an altered bacterial metagenome [10,11]. Similarly, in leptin-deficient mice (*ob/ob*) there is obesity development according to phylum-level changes in the gut microbiome. Specifically, there is a reduced abundance of *Bacteroidetes* and an increased relative abundance of *Firmicutes* [12].

Technical improvements in metagenomics have allowed a better description of the microbial genome and its interaction with the host immune system and the commensal microbiota, providing great benefits, among others, to bench-based and clinical immunology [13]. In detail, gut microbiome and human genome interactions can be studied at the level of transcriptome profiles of innate and adaptive immune cells until epigenetic regulation of cytokine expression. Moreover, we can also study the potential effect of genetic mutations on immune-mediated healthy and pathological conditions [14]. Interestingly, mouse model data support the hypothesis that impairment in human–microbial genomic cross-talk is associated with the pathogenesis of several multi-factorial diseases: infectious diseases, metabolic diseases (e.g., obesity, diabetes, hypertension, metabolic associated fatty liver disease, or MAFLD), inflammatory and autoimmune conditions, aging, cancer and cancerogenesis, and neurodegenerative and neurologic diseases [15,16]. In addition, a multitude of evidence from non-mouse experimental models and one in particular from Cynomolgus monkeys, shows that the study of microbial derived metabolites (namely the “microbial exposome“ that accomplish the gut microbial metabolites present in any fluids or tissues of the host) can greatly affect human metabolism and in particular cancerogenesis process within the colon [17] (Figure 1). 

Therefore, we have strong expectations for metagenomics to help build up personalized genomic-based microbiome-mediated therapeutic strategies. Specifically, we can expect that a double approach can effectively modulate metagenome. First, we could modulate gut microbiota and microbiome, “healing “human genome. Second, we could modulate human genome expression and beneficially affect gut microbiome, positively affecting human health. All this future “personalized medicine” approach requires big data analysis, correlation models of analysis and machine learning support [18].

Thus, we aimed to review literature pieces of evidence on the role of the gut microbiome and its interaction with the human genome studied with newer metagenomics techniques in the frame of health and disease in humans. 

## 2. Materials and Methods

We performed a literature search for topic keywords: gut microbiota, gut microbiome, metagenomics, immune system, and obesity. We also pursued their acronyms and sought keywords association. We mainly used PubMed and Medline search. We included the following types of articles: original research, reviews, meta-analyses, and case series. Abstracts from the main national and international gastroenterological meetings (e.g., United European Gastroenterology Week, Digestive Disease Week) were also included in the research. 

The contributions resulting from the search were reviewed by two of the authors (ES and LB) according to PRISMA guidelines [19]. The last MEDLINE search was dated 31 January 2023.

## 3. Results

### 3.1. Human Genomic Control by the Microbiome

This section of results describes the concept of gut microbial influence on human genome and first recognized mechanism involved.

The microbiota are ubiquitous. The SARS-CoV-2 pandemic has attracted our attention to the unrealistic dream of total sterility of our body. In fact, despite the heavy and frequent use of personal protection equipment, it is almost impossible to avoid virus contact [20]. Thus, we must accept our co-existence with the microbiota throughout our life. Considering such close interaction, it is logical to assume that microbiota influence host gene expression in all body areas that it colonizes, such as skin, respiratory organs, gastrointestinal, and urogenital tract. Two main mechanisms allow the microbiota to “regulate” our genome: through direct body exposure to microorganisms [21] or those of their metabolites [18,22]. Indeed, we must recognize a fine distinction between microbial antigen-mediated genome modulation and metabolite-mediated one [19]. 

The hypothesis and idea that commensal microbiota could modulate both organization and regulation of expression of the human genome started about a decade ago. Intestinal biopsies using brand-new DNA microarrays discovered significant differences in gene expression between germ-free mice and mice colonized with commensal microbiota [23]. Colonization with a single commensal bacterial strain was proven to be sufficient to affect the expression of genes involved in several physiological processes (e.g., nutrient metabolism, tissue differentiation, and immune system activity) [19,20]. Moreover, the introduction of a single enteric viral strain into germ-free animals dramatically changes the gene expression profile of the enterocytes, albeit maintaining their physiological functions [24]. Moreover, introducing entire microbial communities in the gut of germ-free mice results in massive and very complex (cell type-specific) transcriptional responses [25]. Similarly, upon microbial colonization after birth, intestinal gene expression undergoes dramatic reprogramming, which is partially dependent on microbial sensing receptors of the innate immune system [22,26]. Physiologically, these experimental observations mimic the recognized and not yet completely described regulatory effects on our genome of successive stages of microbial colonization of our gut occurring after birth. They are, of course, a determinant part of our intestinal system maturation (e.g., gut-associated lymphoid tissue, or GALT) [27]. Thus, the metagenomics description of metagenome resembles the notion of a “superorganism” determined by the fine interaction between eukaryotic and prokaryotic genome cross-regulation [28]. 

The transcriptional response resulting from intestinal microbial colonization seems to have species-specific characteristics. Furthermore, this peculiarity seems to work for some species. For example, gene regulation differs between mice and zebrafish whenever microbiota transplantation is operated [29].

Despite metagenomics techniques allowing significant progresses in the understanding of the influence of intestinal microbes on host gene regulation, the mechanisms involved in transcriptional reprogramming remain largely unknown [30,31] (Figure 2). 

The first mechanistic hints of this functioning were obtained from studies on the effects of the host epigenome on microbiota [32]. In particular, the methylation levels of the gene encoding Toll-like receptor 4 (Tlr4) were lower in germ-free mice compared to those colonized by commensal bacteria [33]. Furthermore, mice with a conditional deletion of the *histone deacetylase 3* (*Hdac3*) gene in intestinal epithelial cells presented intestinal barrier function derangements (e.g., depletion of Paneth cells, and consequent high frequency of intestinal inflammation [34]). These phenotypic changes within the intestine depend on microbiota, because germ-free mice with the same deletion do not have them. Moreover, when transferring the microbiota of *Hdac3*-deficient mice to the intestine of Hdac3-sufficient germ-free mice, the pathologic phenotype was not observed. 

Importantly, Camp et al. first showed that microbial modulation on intestinal gene expression occurs independently of the spatial organization of nucleosome-depleted accessible chromatin [35]. The chromatin accessibility landscape of germ-free mice was similar to that of conventionally raised mice. Furthermore, the same trend was also observed in germ-free mice transplanted with normal commensal gut microbiota [32]. 

Altogether, these findings suggest that commensal bacteria regulation of the intestinal genome has a “multi-hit” shape affecting transcription factor binding to open chromatin. Deep characterization of these signaling events may help in understanding the ways in which host (e.g., human) tissues respond appropriately to microbial colonization through transcriptome modifications.

The mechanistic interaction of non-genomic processes mediated by the microbiota and epithelial cell processes, such as glycosylation and cargo sorting, can be integrated into the larger epigenetic influence of the gut microbiome on the intestinal genome [36,37] (Figure 2).

A special mention is dedicated to the impact of gut microbiome on our immune system processes. In fact, gut microbial colonization can affect gene expression in immune system cells [38]. Four days from gut microbial colonization of the intestine of germ-free mice, major transcriptional induction of innate and adaptive immune genes occurs. The latter includes expression of anti-microbial peptides, lineage transcription factors of T cells and molecules involved in antigen presentation [38]. Further, myeloid cells of the intestinal mucosa are the very first line cells of innate immunity, and they show rapid transcriptional responses vs. microbial colonization. Thus, they are characterized by induction of genes involved in the inflammatory response (e.g., genes encoding type I interferons in intestinal mononuclear phagocytes) [39]. On the other hand, microbial-produced butyrate downregulates this pro-inflammatory gene expression in intestinal macrophages [40]. Interestingly, short-chain fatty acids also have a transcriptional control in regulatory T cells. Specifically, butyrate increases the number of peripheral regulatory T cells through inhibition of histone deacetylation in intronic enhancer sequences of the *FoxP3 locus* [41]. This can explain the improvement of T-cell-dependent colitis development in mice upon butyrate administration. 

Another T-cell subset strongly influenced by the microbiota is the T helper 17 (Th17). However, this influence seems to be reserved to those helpers localized within the lamina propria. Therefore, we can hypothesize that gut microbiota effects on genes of host immune system cells also follow a biogeographical localization [14]. 

Gamma–delta T cells and natural killer T (NKT) cells are another “genomic target“ for gut microbiota. These cells express T-cell receptor but also mediate innate immunity functions (e.g., rapid cytokine production). Indeed, gut microbiota can reprogram the transcriptome of intraepithelial γδ-T cells. In fact, commensal microbiota colonization of mice during their neonatal period is associated with decreased CpG methylation in the 5’ region of the gene encoding the chemokine CXCL16. The resulting reduced Cxcl16 expression is able to protect pulmonary and intestinal mucosa of mice from increased mucosal accumulation of NKT [42]. This evidence indicates that the regulatory effect of the microbiota on host immunogenomics and epigenomics is not limited to the gut but is a pleiotropic process. 

### 3.2. Genomic Control of Gut Microbiome by the Host

This section of results describes human genomic influence on gut microbiota composition, with examples in health and disease.

The question is: How does the gut microbiome “react” to the human genome and impact bacterial communities structuring and composition? Furthermore, what is the effect of this “collision” on human health? [43].

There are essentially two paradigms describing person-specific gut microbiome shuffling: effects of diet and genetics. The effect of ingested food is gaining more and more evidence supporting its strong impact on the gut microbiome. However, this impact is rapid and manifests within days [44]. Thus, the host genome represents a reliable chance to understand these re-shaping mechanisms of the gut microbiome in the frame of the metagenome, with possible future therapeutic implications for human health and disease treatment. 

A systematic investigation of factors affecting gut microbiome and designing the microbial ecosystem showed that host genome determines the diversity of the microbiome in mice [45]. Consequently, in human monozygotic twins (namely those sharing one genotype), the microbiota are significantly similar [46]. Very interestingly, host genomes and environmental factors (e.g., food nutrients) have a significantly different impact on distinct members of the gut microbiota. The abundance of the *Christensenellaceae* taxa is more highly correlated within monozygotic than within dizygotic twins. On the other hand, the abundance of *Bacteroidetes* taxa is mainly conditioned by environmental factors [38,47]. More interestingly, monozygotic twins seem to also share highly concordant gut archaea profiles (e.g., the methanogen *Methanobrevibacter smithii*) compared to dizygotic twins [48]. 

We can therefore infer a strong correlation of certain host genes with the abundance levels of microbial taxa. A study examined the possible linkage existing between the C57BL/6 J inbred mouse strain and an ICR/HaJ-derived outbred line [37]. Eighteen quantitative trait loci (QTLs) of the host presented significant linkage with the abundances of specific microbial taxa. In particular, these host loci control individual microbial species, groups of related taxa, or groups of phylogenetically distant microorganisms. Genes involved in this host genome-driven taxonomic effect on gut microbiome were those responsible for immune signaling (e.g., *Irak3*, *Lyz1*, *Lyz2*, *IFN-gamma*, and *IL-22*) [37]. In another study, a mouse inbred line was used, which is often employed to verify differences in susceptibility to obesity and other metabolic disorders [49]. It was discovered that QTLs influence gut microbial composition [50]. More interestingly, subsequent genome sequencing of the QTL regions allowed the generation of candidate genes potentially responsible for the different gut microbiota quali-/quantitative abundance changes. For example, QTL located on chromosome 15 hosts the candidate gene *Irak4* and has a significant association with *Rikenellaceae* abundance; QTL mapped on chromosome 12 hosts the candidate gene *TGF-beta 3* and affects *Prevotellaceae* abundance. Furthermore, the QTL region on chromosome 4, a region rich in interferon genes, was significantly associated with the diversity and abundance of *Bacteroides* [51]. Altogether, all these data support the strongest associations for immune pathway-related genes [43]. Thus, the immune system seems to be the major causative element in re-shaping the host-specific microbiome. Further, it can be hypothesized that immune genome variations can help explain different gut microbiota representations typical of any individual. This variability can directly relate to inflammatory (bowel and non-bowel) disease. For instance, it has been demonstrated that a clear lack of functional interaction between the human genome and microbiome has a significant pathogenic role in inflammatory bowel disease (IBD) outbreak. The latter has consolidated evidence of gut dysbiosis and alterations in the microbiome [52,53]. Interestingly, several studies aimed to verify the effect of known/candidate risk alleles for IBD on metagenomic stability [54]. For example, the innate immune receptor NOD2 and the autophagy-related protein ATG16L1 has a significant association with changes in the gut microbiome [55,56]. Specifically, the relative abundance of *Faecalibacterium* and *Escherichia* taxa is significantly associated with *NOD2* and *ATG16L1* genotypes and, importantly, disease expression phenotype. Thus, these findings support the hypothesis that both genetic assets and disease phenotypes can affect metagenome [57,58] (Figure 2). 

## 4. Gut Metagenome in Health and Disease

This section of results describes the role of metagenome (human) host physiology and pathophysiology belonging to pathologic conditions. QTL, QTN examples are provided.

There are few but promising pieces of evidence supporting the role of gut metagenome in human health and disease. 

A cutting-edge work by Markowitz et al. showed the triangular association between human genome variations, gut microbiome composition and health and disease predisposition [59]. The authors screened nearly a thousand gut microbiome-associated genetic variants (MAVs) and their impact on phenotypes reported in electronic health records from tens of thousands of human individuals (mainly European). Interestingly, they detected a statistically significant association of several MAVs with neurological, metabolic, digestive, and circulatory system diseases. Furthermore, five MAVs correlated with the relative abundance of microbes in the human intestine. These pathophysiological relationships are independently verified according to data from case-control studies matching microbes by disease [52]. 

In detail, human genomic variability is associated with those of microbiome in several organs and tissues (e.g., the gut, skin, vagina, and mouth) [51]. The latter are significantly associated with gut microbiome-associated variants (MAV). For example, the most common MAVs are those between the lactose digestion LCT/MCM6 genomic region, associated with an abundance of gut genus *Bifidobacterium* [60]. 

Determining whether human genome variation is associated with differential different gut microbiota variants and disease risk in a triangular relationship is a determinant step for personalized medicine. Whenever a gut microbial taxon inherits MAV determined by human physiological or metabolic living behavior, it becomes the target for potential modulation (e.g., lifestyle changes, diet, use of antibiotics, pre-, probiotics) [61].

Technically, through the recognition of MAVs from large and geographically diverse populations of healthy humans, it is feasible to assess the impact of human genetic variation on the microbial genome and taxonomic composition variations. Thus, these data can help understand the regulation of the expression of human genome and the evolutionary origin of MAVs. 

MAVs occur in all 22 pairs of autosomes, in coding and noncoding regions of the genome, including also the expression quantitative trait loci (eQTL). The latter are important because the nucleotide variation is associated with the differential expression of a target gene [62]. Environmental factors can affect gene regulation through eQTLs according to their rate of variation [63]. Moreover, both animal and human studies suggest a significant association between similar microbiome and gene regulation mechanisms. Specifically, inter-hominid species data confirm that microbiomes associated with gene interactions are the most conserved (namely traits for regulation of inflammation and apoptosis) [64]. 

In the investigation by Markowitz et al., 925 MAVs were detected and 908 had annotation according to wide genomic population-based database matching [59]. Interestingly, only 4 of 908 MAVs were protein coding (namely two synonymous and two non-synonymous variants). Further, 437 out of the remaining 904 MAVs were intergenic; 415 were intronic, and, finally, 45 were variants in the 3’ untranslated region (UTR; 18) or the 5’ UTR. 

Analyzing MAV eQTL target genes, they were found in the following tissues: skin, esophagus, thyroid, nerves, arteries, adipose tissue, blood, testis, skeletal muscle, lungs, colon, heart, pancreas, spleen, and pituitary gland. In addition, there was significant overrepresentation shared among 15 enriched tissues (mainly colon, heart, and lung). Genetic common physiologic cascades included interferons synthesis, T-cell receptors, and Programmed Cell Death Ligand 1 (PD-1) signaling. Generally, physiologic and related pathophysiologic processes highlighted by MAVs eQTL target genes analysis described the life-long host–microbiome immunological interaction [52].

Subsequently, PheWAS analysis was performed using the 908 annotated MAVs in populations (with European or African origin). PheWAS is a regression analysis to detect whether a genetic variant is associated with a disease based on large population-based medical databases (irrespective of the phenotype) that have been genotyped at a given locus [52]. Interestingly, 31 clinical traits were associated with 10 MAVs in the European cohort only. More specifically, these variants were present on chromosomes 2, 3, 6, 9, 15, and 18. Interestingly, the largest number of associations originated from three MAVs on chromosome 6 in the human leukocyte antigen (HLA) region. Moreover, clinical traits identified MAVs’ associations belonging to circulatory system, neurological, skin tissues, endocrine/metabolic processes, musculoskeletal, hematopoietic diseases, digestive, neoplasms, and sensitive organs. Importantly, 6 out of 10 MAVs were associated with neurological, hematological, dermatological, and metabolic phenotypes. They were also matched with corresponding eQTLs in brain, vascular, skin, and gastrointestinal tract tissues.

In post hoc analysis, each of the eight replicated phenotypes was matched with human case–control data to detect a specific disease associated with a specific gut microbiota asset. Interestingly, five of eight associations were detected. Every triangular association described human genotype linked with both disease risk and specific gut microbial composition change. Thus, it was demonstrated that the human genome influences pathologic phenotype and microbiome. The latter affects or is affected by diseases. These physiopathological “triads” include the core gut family *Lachnospiraceae* [65]. Indeed, *Lachnospiraceae* are involved in a process of diet-derived polysaccharides degradation. Moreover, this microbial abundance is significantly associated with inflammatory conditions, depressive syndromes, and, last but not least, multiple sclerosis.

Another example of the relationship between MAVs, gut dysbiosis, and disease is rs9357092 (G), associated with an increased risk of multiple sclerosis and reduced abundance of *Coriobacteriaceae* family, a commensal bug of the oral, gut, and genital microbiota [66]. In particular, this bug is depleted in the guts of untreated multiple sclerosis patients [67]. In detail, this MAV is located within a zinc ribbon domain containing the pseudogene *ZNRD1ASP*, in close proximity to the HLA complex. Conversely, rs11751024 (C) correlates with the decreased relative abundance of the bacterial family *Lachnospiraceae*. This MAV is co-abundant in the gene group involved in relapsing–remitting multiple sclerosis in adult [68] and pediatric patients [69]. This intergenic MAV is located within the HLA genomic complex, namely between HLA-DQA1/HLA-DRB1 and HLA-DRB5/HLA-DRB9. However, MAV rs11751024 has other additional clinical associations: psoriasis, celiac disease, and type 1 diabetes. Thus, an increased risk for cardiovascular disease and diabetes is associated with a decreased concentration of Lachnospiraceae (CAG-882) [70,71] (Table 1).

There are also intestinal MAV phenotypes associated with hematological and cardiovascular disease risk for pulmonary embolism and infarction, venous embolism and thrombosis, and pulmonary heart disease. Specifically, these pathologic features are associated with intronic MAVs rs8176645 (A) and rs3758348 (C) of ABO and SURF4, respectively. Importantly, the ABO/SURF4 region is one of a few recurring genomic regions linked to the gut microbial composition and associated with cardiovascular risk [72]. It is important to note that rs8176645(A) (ABO risk allele) presence is associated with a reduced abundance of *Bifidobacterium bifidum*. On the other hand, rs3758348 (C) (SURF4 risk allele) is associated with an increased abundance of *Faecalibacterium* [73]. From a physiologic point of view, the SURF4 risk allele is significantly over-expressed and associated with increased blood protein levels of platelet endothelial cell adhesion molecule-1, independently associated with thrombosis [74]. Therefore, we can hypothesize that the MAV-induced increase in the abundance of *Faecalibacterium*, whose butyrate metabolites have anti-thrombotic effects “compensate” MAV linked to cardiovascular disease risk increase. Accordingly, *Faecalibacterium* is depleted in older patients with coronary artery disease and heart failure [75] (Table 1).

Only a specific single association between MAV and metabolic disease was detected, namely gout and MAV rs3749147 (A). This association was correlated with the increased relative abundance of *Eggerthella* [52]. Specifically, the gout-linked MAV is an eQTL targeting GCKR, a glucokinase inhibitor typical of gout [52]. However, *Eggerthella* does not have a greater abundance in the urine of gout patients [52,76] (Table 1).

Metagenomics Data Are in Line with the Study of MAV

In the frame of parallel pathophysiology linking immune dysregulation and disorders of metabolism, we can detect a metagenomic linkage between inflammatory bowel disease and type 2 diabetes/obesity. There is a similar decrease in microbial species and gene expression diversity in obesity [77]. Specifically, there is a reduction in the *Firmicutes* and *Clostridia* abundance and an increase in the *Firmicutes*-to-*Bacteroidetes* ratio [78]. More interestingly, *Faecalibacterium prausnitzii* is as reduced in abundance as in type 2 diabetes patients [72]. From a metagenomic point of view, these patients show increased gene expression for those involved in membrane transport, sulfate reduction, and resistance to oxidative stress. On the other hand, there is decreased expression of genes for cofactors, vitamin metabolism, and butyrate production [79]. 

MiRNA is an endogenous small noncoding RNA molecule, with 18 to 25 nucleotides. It is able to regulate gene expression through degradation of mRNAs or inhibition of its translation. MiRNA regulates cell differentiation, proliferation, and, subsequently, tumorigenesis, and also immune system functioning [80]. Interestingly, MiRNAs is also involved in the modulation of the commensal microbiota-dependent intestinal epithelial cells, present in the maintenance of eubiosis vs. dysbiosis [81]. Thus, miRNAs dysfunction can result in cancer development and autoimmune disorders [82].

As gut microbiotas’ metabolites are effective host metabolic regulators and are affected by epigenetic mechanisms, the latter could modulate host metabolism. Interestingly, there is interaction between miRNA and gut microbiota in obese patients [83]. Twenty-six different miRNAs and 12 microbial species showed a significant correlation in obese subjects. In particular, three miRNAs (namely miR-130b-3p, miR-185-5p, and miR-21-5p) inversely correlated with *Bacteroides eggerthii* concentration. More interestingly, these miRNAs had a BMI-regulating impact. In addition, the expressions of miR-107, miR-103a-3p, miR-222-3p, and miR-142-5p was inversely correlated with *B. intestinihominis* abundance. More specifically, these miRNAs regulate genes involved in fatty acid degradation, insulin signaling, and glycerol lipid metabolism. Moreover, miR-15a promotes insulin biosynthesis through inhibition of the expression of endogenous uncoupling protein 2 (UCP2). This results in increased levels of ATP and glucose-stimulated insulin secretion. Finally, through big data analysis, 14 circulating miRNAs (miR-107, miR-103a-3p, miR-142-5p, miR-222-3p, miR-221-3p, miR-183-5p, miR-130b-3p, miR-15a-5p, miR-33a-5p, miR-210-3p, miR-144-3p, miR-185-5p, miR-130a-3p, and miR-21-5p) expression rate and abundance of four intestinal microbial taxa (namely *D. longicatena, B. intestinihominis, B. eggerthii, and H. parainfluenzae*) were determined to be significantly higher in obese subjects, with a peculiar correlation behavior [78].

## 5. Conclusions and Future Directions

The epigenetic impact of the gut microbiome on the human genome is largely considered the first hint of the existence of gut metagenome. Transcriptional regulation can be mediated by metabolites produced by intestinal microbes or through microbe-immune interactions. Non-epigenetic mechanisms mediated by the gut microbes can also affect the expression of our genes. 

At the same time, the human host genome can affect gut microbiome asset in health and disease. IBD is a significant example of the translational potential of reprogramming the metagenome in the frame of personalized medicine due to the pivotal role of AI modeling (namely big algorithms for big data analysis). 

In the perspective of “personalized medicine“, the modulation of human gut microbiota is one way we can hypothesize the modulation of metagenome. This process would start from pre-, pro-, postbiotics, fecal microbiota transplantation and antibiotic use and would imply both transcriptional regulation and non-epigenetic mechanisms. On the other hand, human genome reprogramming through “personalized“ lifestyle, diet and other host-related factor changes can be the beginning of the future therapies impacting gut metagenome. 

## Figures and Tables

**Figure 1 genes-14-00857-f001:**
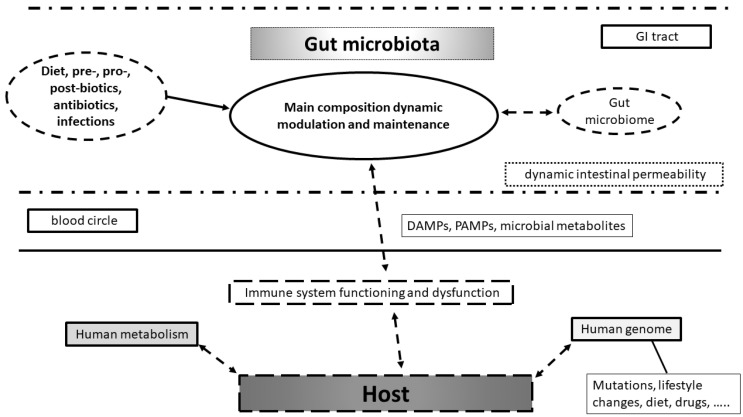
Model of interaction between environment, gut microbiota, microbiome and host with its genome in health and disease. Diet, pre-, pro-, postbiotics, antibiotics are used, but disease can dynamically affect gut microbiota composition and microbiome asset. These changes can modulate immune system functioning in bi-directional manner. Immune system modulation can affect host metabolism in health and disease with human genome changes. These are already related to mutations occurrence, lifestyles changes, diet, and drug use.

**Figure 2 genes-14-00857-f002:**
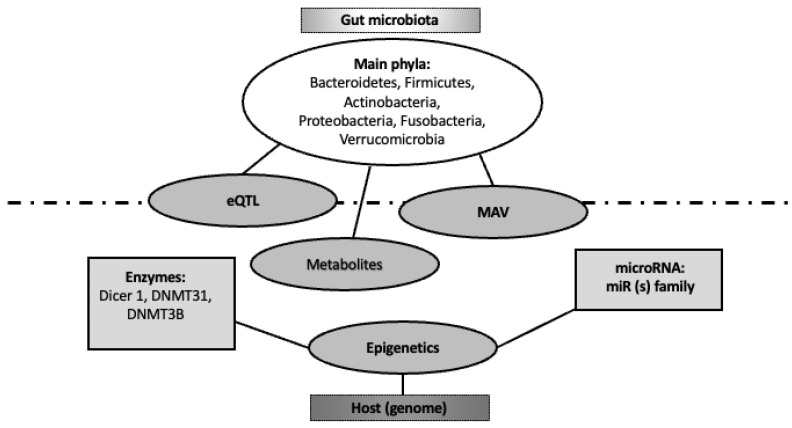
Bi-directional interaction between gut microbiome and human genome. The host (genome) regulates gut microbiota composition and its microbiome through several epigenetic items (namely enzymes and miRNAs). On the other hand, gut microbiome modulates host genome through its metabolites, targeting expression quantitative trait loci (eQTL) and microbiome-associated genetic variants (MAVs).

**Table 1 genes-14-00857-t001:** MAV, gut microbial association and disease significance.

*MAV*	*Gut Microbial Association*	*Associated Pathologic Expression*
rs3749147	*Eggerthella*	Gout
rs11751024	*Lachnospiraceae (CAG-882)*	Type 1 diabetes
rs11751024	*Lachnospiraceae (CAG-882)*	Celiac disease
rs11751024 rs9357092	*Lachnospiraceae (CAG-882)* *Coriobacteriaceae (f)*	Multiple sclerosis
rs3758348	*Faecalibacterium* *Bifidobacterium bifidum*	Deep vein thrombosis, chronic pulmonary heart failure
rs11751024	*Lachnospiraceae (CAG-882)*	Psoriasis

## Data Availability

All the data reviewed in this manuscript can be found online on PubMed, Medline and database from main national and international Gastroenterology congress.

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
