# Peer review of "Human Genes Involved in the Interaction between Host and Gut Microbiome: Regulation and Pathogenic Mechanisms"

_genes, 2023, doi:10.3390/genes14040857_

Round 1

Reviewer 1 Report

The authors have tried to review human genome interactions and its regulatory role with existing human metagenome. There are few reviews already published such as:

https://www.ncbi.nlm.nih.gov/pmc/articles/PMC7680557/

https://www.ncbi.nlm.nih.gov/pmc/articles/PMC5774629/

https://www.pnas.org/doi/10.1073/pnas.2200551119

Could you please define novelty in this review with respect to these published reviews?

The structure of review needs to be modified as QTLs are discussed in two separate sections of the review and it is not defined, either based on disease or genome elements. It is little scattered and poorly written. The language of the review also needs to be improved.

Author Response

REPLY POINT BY POINT TO REVIWERS’ QUERIES

REVIEWER 1:

The authors have tried to review human genome interactions and its regulatory role with existing human metagenome. There are few reviews already published such as:

https://www.ncbi.nlm.nih.gov/pmc/articles/PMC7680557/

https://www.ncbi.nlm.nih.gov/pmc/articles/PMC5774629/

https://www.pnas.org/doi/10.1073/pnas.2200551119

Could you please define novelty in this review with respect to these published reviews?

We thank the reviewer for this observation. Our review has combined genome-based observations from literature and, in detail, the point of view of genetics expert (namely, L.B.) and those of clinical expert (namely, E.S.).

The structure of review needs to be modified as QTLs are discussed in two separate sections of the review and it is not defined, either based on disease or genome elements. It is little scattered and poorly written.

We thank the reviewer for this observation. QTL data are discussed on two separate section in order to differentiate the impact of gut microbiota on these and, on the other hand, the impact of QTL changes on gut microbiome and microbiota. We have specified better this distinction within the text.

The language of the review also needs to be improved.

We have performed further accurate revision of the language used.

Reviewer 2 Report

Gut microbiome plays a very important role in our health by helping control digestion, benefiting our immune system and many other aspects of our health.Despite the growing knowledge surrounding host–microbiome interactions, we are just beginning to understand how the gut microbiome influences and is influenced by the host gene expression. Scientific work with model organisms demonstrate that the gut microbiome is an important regulator of different host pathways including immune development, metabolism, and could remodel host chromatin, causes differential splicing, alters the epigenetic background, and directly interrupts host signaling cascades.
The title is informative enough to draw its potential reader's attention. The introduction of this article successfully explains why the bi-directional role of gut microbiome and human genome in host metabolism and immune system regulation is important for human health.
In the introduction/background section of this article, is not so successfully explained why current research is important. But the authors could provide some small additional information related to the problem and include additional relevant references.
Here are some minor suggestions that would improve the introduction:
1.    According to the context in the Introduction, the authors can give a scheme/figure of the interactions between microbiota/microbiota disturbances, factors affecting microbiota, genetic sucseptability, immune dysregulation and diseases.
2.    You can give some information about the association between microbiome and gene expression in other model organisms (except mice).

In chapter “Materials and Methods” - Which guided the selection of articles: genes involved in host-gut microbiome interaction or pathogenic mechanisms that are associated with a disturbed microbiota? Why did you choose obesity/digestive diseases?

In chapter “Results”, according to the context the author could explain briefly the immune response triggered by changes in the gut microbiome and genes involved in the process. The figure 1 needs to be corrected, the letters are not very clear.

In chapter “Gut metagenome in health and disease”, the authors could explain and/or give a figure/scheme for major factors in configuring and influencing the gut microbiome (diet, geography, ethnicity, lifestyle, genetics….). In this chapter, the authors explain very well the data from Markowitz et al. article, but they can represent a table to summarize the data (MAVs and associated diseases/disorders).

The “Conclusions and future directions” chapter can be supplemented by an authors’ proposal how gut microbiome-oriented therapy and host-directed therapy could be personalized in clinical practice using the data from these analyses.

Author Response

REPLY POINT BY POINT TO REVIWERS’ QUERIES

REVIEWER 2:

Gut microbiome plays a very important role in our health by helping control digestion, benefiting our immune system and many other aspects of our health.Despite the growing knowledge surrounding host–microbiome interactions, we are just beginning to understand how the gut microbiome influences and is influenced by the host gene expression. Scientific work with model organisms demonstrate that the gut microbiome is an important regulator of different host pathways including immune development, metabolism, and could remodel host chromatin, causes differential splicing, alters the epigenetic background, and directly interrupts host signaling cascades. 
The title is informative enough to draw its potential reader's attention. The introduction of this article successfully explains why the bi-directional role of gut microbiome and human genome in host metabolism and immune system regulation is important for human health.

In the introduction/background section of this article, is not so successfully explained why current research is important. But the authors could provide some small additional information related to the problem and include additional relevant references.

We thank the reviewer for this observation. We have added the required information and reff.

Here are some minor suggestions that would improve the introduction:
1.    According to the context in the Introduction, the authors can give a scheme/figure of the interactions between microbiota/microbiota disturbances, factors affecting microbiota, genetic sucseptability, immune dysregulation and diseases.

We thank the reviewer for the suggestion. We have added an explicative figure.

  1. You can give some information about the association between microbiome and gene expression in other model organisms (except mice).

We thank the reviewer for these precious informations. We have updated the Introduction section accordingly.

In chapter “Materials and Methods” - Which guided the selection of articles: genes involved in host-gut microbiome interaction or pathogenic mechanisms that are associated with a disturbed microbiota? Why did you choose obesity/digestive diseases?

We thank the reviewer for this useful observation.  We first looked at articles on host-gut microbiome interaction. Then, we found examples of genes involved in regulation of host health and disease manifestations such as obesity and digestive diseases. The latter were significantly associated with gut dysbiosis.

In chapter “Results”, according to the context the author could explain briefly the immune response triggered by changes in the gut microbiome and genes involved in the process. The figure 1 needs to be corrected, the letters are not very clear. 

We thank the reviewer for this suggestion. We have added the suggested data.

We have improved the figure 1.

In chapter “Gut metagenome in health and disease”, the authors could explain and/or give a figure/scheme for major factors in configuring and influencing the gut microbiome (diet, geography, ethnicity, lifestyle, genetics….). In this chapter, the authors explain very well the data from Markowitz et al. article, but they can represent a table to summarize the data (MAVs and associated diseases/disorders).

This is a useful observation. In fact, ee have already added to the introduction this explicative figure.

This is another useful suggestion. We have added this table to the paper.

The “Conclusions and future directions” chapter can be supplemented by an authors’ proposal how gut microbiome-oriented therapy and host-directed therapy could be personalized in clinical practice using the data from these analyses.

We thank the reviewer for this suggestion. We have added the proposals according to this interesting input.

Round 2

Reviewer 2 Report

Dear Authors,

I have been carefully reviewed your revised article with the id number
"genes-227239"In my opinion, this revised article incorporates all of the points raised in the original draft to the best of my knowledge.
The title is informative and relevant. The aim is also stated clear.
The references are relevant and recent. The cited sources are referenced correctly. Appropriate studies are included.The results is presented in an appropriate way.
Tables and figures are made appropriately.
Best wishes to all of the authors who contributed to the production of this wonderful work and congratulations on their future endeavors.